# Incidence and Predictors for Oncologic Etiologies in Chinese Children with Pituitary Stalk Thickening

**DOI:** 10.3390/cancers15153935

**Published:** 2023-08-02

**Authors:** Mario W. T. Li, Sarah W. Y. Poon, Claudia Cheung, Chris K. C. Wong, Matthew M. K. Shing, Terry T. W. Chow, Samantha L. K. Lee, Gloria S. W. Pang, Elaine Y. W. Kwan, Grace W. K. Poon, Ho-Chung Yau, Joanna Y. L. Tung, Anthony P. Y. Liu

**Affiliations:** 1Department of Paediatrics and Adolescent Medicine, The Hong Kong Children’s Hospital, Hong Kong, China; lwt724@ha.org.hk (M.W.T.L.); samanthalklee@hkbh.org.hk (S.L.K.L.); tyl404@ha.org.hk (J.Y.L.T.); 2Department of Paediatrics and Adolescent Medicine, Queen Mary Hospital, Hong Kong, China; 3Department of Radiology, The Hong Kong Children’s Hospital, Hong Kong, China; 4Department of Paediatrics, Prince of Wales Hospital, Hong Kong, China; 5Department of Paediatrics and Adolescent Medicine, School of Clinical Medicine, The University of Hong Kong, Hong Kong, China

**Keywords:** thickened pituitary stalk, central diabetes insipidus, endocrinopathy, germ cell tumor, Langerhans cell histiocytosis

## Abstract

**Simple Summary:**

This is a review of the clinical course in 28 Chinese children with pituitary stalk thickening (PST) and endocrinopathies revealed a higher rate of germ cell tumors (GCT) compared to the literature based largely on non-Asian cohorts. Predictors for oncologic etiologies included male sex, severe stalk thickening ≥6.5 mm, central diabetes insipidus or ≥2 pituitary hormone deficiencies as presenting features. Management guidelines for PST should take into account patient ethnicity with a lower threshold of biopsy for Asian children. Multidisciplinary approach with serial monitoring of clinical, tumour markers, endocrinopathies and imaging are essential for diagnosis.

**Abstract:**

Background: With the increasing use of magnetic resonance imaging (MRI) in the evaluation of children with endocrine disorders, pituitary stalk thickening (PST) poses a clinical conundrum due to the potential for underlying neoplasms and challenges in obtaining a tissue biopsy. The existing literature suggests Langerhans cell histiocytosis (LCH) to be the commonest (16%) oncologic cause for PST, followed by germ cell tumors (GCTs, 13%) (CCLG 2021). As the cancer epidemiology varies according to ethnicity, we present herein the incidence and predictors for oncologic etiologies in Hong Kong Chinese children with PST. Methods: Based on a territory-wide electronic database, we reviewed patients aged < 19 years who presented to three referral centers with endocrinopathies between 2010 and 2022. Records for patients who underwent at least one MRI brain/pituitary were examined (*n* = 1670): those with PST (stalk thickness ≥ 3 mm) were included, while patients with pre-existing cancer, other CNS and extra-CNS disease foci that were diagnostic of the underlying condition were excluded. Results: Twenty-eight patients (M:F = 10:18) were identified. The median age at diagnosis of PST was 10.9 years (range: 3.8–16.5), with central diabetes insipidus (CDI) and growth hormone deficiency (GHD) being the most frequent presenting endocrine disorders. At a median follow-up of 4.8 years, oncologic diagnoses were made in 14 patients (50%), including 13 GCTs (46%; germinoma = 11, non-germinoma = 2) and one LCH (4%). Among patients with GCTs, 10 were diagnosed based on histology, two by abnormal tumor markers and one by a combination of histology and tumor markers. Three patients with germinoma were initially misdiagnosed as hypophysitis/LCH. The cumulative incidence of oncologic diagnoses was significantly higher in boys and patients with PST at presentation ≥6.5 mm, CDI or ≥2 pituitary hormone deficiencies at presentation and evolving hypopituitarism (all *p* < 0.05 by log-rank). Conclusions: A higher rate of GCTs was observed in Chinese children with endocrinopathy and isolated PST. The predictors identified in this study may guide healthcare providers in Asia in clinical decision making. Serial measurement of tumor markers is essential in management.

## 1. Introduction

The pituitary stalk connects the median eminence of the hypothalamus to the pituitary gland [1]. Pathologies involving the pituitary stalk may present with hormonal deficiencies, precocity and non-specific symptoms such as headache and visual disturbance [2,3,4,5,6]. Isolated pituitary stalk thickening (PST) may be a result of neoplasms, such as germ cell tumors (GCTs) and Langerhans cell histiocytosis (LCH), inflammation, infection or congenital variants or can be idiopathic [6,7,8,9,10,11,12,13,14,15]. With the increasing use of magnetic resonance imaging (MRI) in the evaluation of probable growth and hormonal disturbances, PST is increasingly being diagnosed in children and adolescents [6,16,17,18,19]. Given the potential morbidities and concerns regarding sampling errors in surgical biopsies, the management of children with PST represents a clinical conundrum for physicians [20,21,22,23,24,25,26]. Risk assessment by clinical, biochemical and radiographic features is essential in making an informed decision in order to avoid unnecessary surgical procedures and delays in tumor diagnosis.

The National Institute for Health and Care Excellence (NICE) has formulated guidelines on the management of children and adolescents with PST and/or central diabetes insipidus (CDI) with no apparent diagnosis [27]. A review of the literature indicates that 45.5% of stalk thickenings are due to neoplastic causes, while 29.1% are idiopathic. Specifically, LCH is the most common oncologic diagnosis (16.1%), followed by GCT (13.1%) and craniopharyngioma (12.3%). Congenital lesions account for 19.1%, with the commonest being septo-optic dysplasia (5.8%). Only a minority are found to have infectious, inflammatory, autoimmune or post-traumatic etiologies.

Nonetheless, it is well established that the epidemiology of CNS tumors varies according to ethnic groups. In particular, CNS GCTs have a significantly higher incidence in Asians when compared to the Western population, which the current literature is largely based on [28,29]. In addition, report cohorts do not precisely address the question of isolated PST, and pathologies involving the hypothalamus (such as craniopharyngioma) and pituitary gland (such as pituitary adenoma) are frequently included. Herein, we report the incidence rates, risk factors and predictors for neoplasms in a multi-institutional cohort of Chinese children with PST presenting with endocrinopathies in Hong Kong. Our findings are discussed in relation to the applicability of the NICE guidelines for our local population. As an exploratory study, the feasibility of incorporating 11C-methionine PET-MRI in the context of PST is also evaluated.

## 2. Materials and Methods

### 2.1. Study Cohort

This was a retrospective cohort study of Chinese children presenting with endocrinopathies and the MRI findings of PST, managed at two university-affiliated pediatric units (Queen Mary Hospital, Prince of Wales Hospital) and Hong Kong Children’s Hospital between 2010 and 2020. Ethics approval was granted by the Hospital Authority Central Institutional Review Board (reference number: PAED-2023-009).

Study subjects were identified using the territory-wide electronic patient database Clinical Data Analysis and Reporting System (CDARS). The database was first inquired for pediatric patients (<19 years of age) diagnosed with one or more endocrinopathies, including growth hormone deficiency (GHD), hypogonadotropic hypogonadism (HH), central hypothyroidism (CH), adrenal insufficiency (AI), central precocious puberty (CPP) and central diabetes insipidus (CDI) (ICD 9CM codes 253.2, 253.3, 253.4, 253.5, 255.4, 259.1). The patient list was further filtered for patients who had at least one MRI of the brain and/or pituitary region performed in the three study institutions. Individual clinical records (*n* = 1670) were then reviewed to identify patients according to the following criteria. Inclusion criteria included (1) Chinese patients <19 years of age at first imaging, (2) neuro-imaging performed as part of evaluation for the presenting endocrinopathies, (3) PST according to criteria specified in the NICE guidelines [27] and (4) follow-up (FU) duration of 2 years or more in patients without oncologic diagnosis. Exclusion criteria included (1) patients with radiographic abnormalities beyond PST, such as mass lesions within the pituitary gland or the hypothalamic region, (2) patients with known oncologic diagnosis prior to the index neuroimaging (3) and patients with extra-CNS features that were diagnostic of the underlying condition—for example, skeletal or cutaneous features in LCH.

### 2.2. Data Collection

We reviewed individual patient records for demographic, clinical, biochemical, histological and outcome data. These included sex, age at presentation, age at diagnosis of PST, symptoms and hormonal profiles at diagnosis of PST, before treatment and during FU, available tumor marker levels (serum and cerebrospinal fluid (CSF) alpha-fetoprotein (AFP) and human chorionic gonadotropin (hCG)), extent of PST, morphologic characteristics of pituitary stalks and histologic features on sampled tissue. In patients with subsequent oncologic diagnoses, details of the anti-tumor therapy and treatment outcomes were captured.

### 2.3. Definition of PST

As the designation of PST varies in the literature [28,30,31,32,33,34], we adopted definitions from the NICE guidelines in this study [27]. MRI images for the included subjects, where available (*n* = 24), were reviewed by two radiologists (C.C. and C.K.C.W.). For patients presenting with clinical features (as in our cohort), the cut-off for PST was 2 mm at the pituitary insertion or 3 mm at the level of the optic chiasm. The degree of PST was classified as mild (<4 mm), moderate (4–6.5 mm) and severe (≥6.5 mm) thickening [18].

### 2.4. Primary and Secondary Outcomes

Our primary outcome was the cumulative incidence rate of oncologic diagnosis in the study cohort. The secondary outcomes were predictors for underlying oncologic etiologies and the natural course of endocrinopathy in patients with or without a defined neoplastic diagnosis.

### 2.5. 11C-Methionine PET-MRI

The 11C-methionine (MET) PET-MRI studies were performed in a PET-MR scanner with simultaneous MR DIXON HiRes sequences for attenuation correction and T1 3D MPRAGE for localization. Amino acid PET brain studies were performed at 25 min after the administration of 0.2 mCi/kg of 11C-methionine. The PET uptake images were presented in axial, coronal and sagittal planes. The lesion-to-normal region SUVmean and SUVmax ratios (LNRmean and LNRmax) were determined with reference to the normal contralateral corresponding structure. For the MR study, the following sequences were obtained: sagittal GRE T1-weighted with and without fat suppression; axial SE T1-weighted; axial Turbo SE T2-weighted; axial diffusion-weighted with ADC map; axial SWI; axial FLAIR; coronal FLAIR; sagittal turbo SE T2-weighted pituitary gland; post-contrast sagittal 3D vibe; and space T1-weighted with fat suppression.

### 2.6. Statistical Analysis

Continuous data were presented as medians with ranges, whereas categorical data were presented as counts with percentages. Numerical variables were compared between groups by the Mann–Whitney U test and categorical variables were compared between groups by the Chi-square or Fisher’s exact test. The cumulative incidence of oncologic events was derived from Kaplan–Meier estimates, with the date of diagnosis of PST defined as date of the first MRI revealing such a finding, the date of the event defined as the date of biopsy in those who underwent tissue sampling supporting the final cancer diagnosis or the date of first tumor marker elevation. Patients without an oncologic diagnosis were censored on the date of their last FU. Comparison of the cumulative incidence between groups was performed using a log-rank test, as no competing outcome was identified. Hazard ratios (HR) were obtained with Cox regression. A *p* value < 0.05 was considered statistically significant.

## 3. Results

### 3.1. Demographics and Baseline Characteristics

Twenty-eight patients were identified for our study, including 18 females (64%) and with a median age of diagnosis of PST of 10.9 years (range: 3.8–16.5) (Appendix A). The initial symptoms were polyuria and polydipsia (*n* = 17, symptom duration median = 3 months, range 1–35 months), growth retardation (*n* = 9), precocious puberty (*n* = 8) and delayed puberty (*n* = 5). These were attributed to biochemically confirmed endocrinopathies, including CDI (*n* = 18), GHD (*n* = 14), CPP (*n* = 7), AI (*n* = 7), HH (*n* = 7) and CH (*n* = 6). Fifteen patients (54%) had ≥2 hormonal deficiencies at diagnosis of PST. Visual problems represented the second most common class of presenting symptoms in these patients. Among the nine patients (32%) with visual issues, four had strabismus, two had visual field defects, one had papilloedema and one had amblyopia.

### 3.2. Etiology for PST and Patient Outcome

With a median FU of 4.8 years (range: 0.5–19.3), 14 patients (50%) were diagnosed with oncologic conditions, namely GCT in 13 (46%; germinoma = 11; non-germinomatous GCT (NGGCT) = 2) and LCH in one (4%) (Figure 1). The diagnoses were made after tissue biopsy in 10, the detection of elevated tumor markers in two (both in serum and CSF) and a combination in one. The patient with LCH was diagnosed clinically based on normal tumor markers and the response to the LCH treatment regimen (FU for 9.4 years).

Three patients with GCTs were previously misdiagnosed as having hypophysitis (*n* = 2) or LCH (*n* = 1). Patient No. 8 had an initial diagnosis of hypophysitis and underwent three biopsies over the course of 22 months before the diagnosis of an embryonal carcinoma was made. The first biopsy contained normal pituitary tissue, while the second biopsy contained tissue with inflammatory infiltrates that was considered to be indicative of hypophysitis. His blood and CSF AFP and HCG were normal initially but the serum HCG level was raised to 7 IU/L (normal < 5 IU/L) shortly after the third biopsy. Patient No. 13 was initially managed as LCH with steroid and vinblastine. He was eventually diagnosed with germinoma 3 years later, when he progressed with ventricular metastasis. His serum and CSF AFP and HCG were normal initially but the serum and CSF HCG became raised 3 years later when he was diagnosed with ventricular metastasis. Patient No. 1 presented with decreased growth velocity and CDI, which later evolved into panhypopituitarism with PST of 5 mm. Her serum and CSF tumor markers were again all normal. A biopsy was performed, and she was initially misdiagnosed as having lymphocytic hypophysitis. However, with a further pathology review, germinoma cells were identified, once again highlighting the caveats for diagnosis with the availability of tumor tissue. The median time from the appearance of the first symptom to the diagnosis of GCT/LCH was 1.3 years (range: 0.5–5.4). The cause of PST in the rest of the patients (*n* = 14, 50%; median FU 3 years, range: 1.3–12.2) remained idiopathic. Among the study cohort, one patient with NGGCT died due to disease progression.

### 3.3. Risk Factors for Underlying Neoplasm

Figure 1 shows the cumulative incidence of neoplasms based on clinical, biochemical and radiographic risk factors. Male gender, central diabetes insipidus, ≥2 hormonal deficiencies at diagnosis of PST and PST ≥ 6.5 mm at presentation were predictors of oncologic etiologies. In addition, the evolution of the endocrinopathies was significantly associated with underlying neoplasms.

### 3.4. Clinical Course and Endocrine Outcome

Table 1 summarizes the evolution of the endocrinopathies during the study period. At the last FU, CDI remained as the most common endocrinopathy (*n* = 18), followed by GHD (*n* = 14), AI (*n* = 12) and CH (*n* = 12).

For the patients with an oncologic diagnosis, the evolution of a pituitary hormone deficiency was more common (*n* = 9/14; 64%) than in the idiopathic group (*n* = 2/14, 14%) (evolution of pituitary hormone deficiencies was more common in patients with oncologic diagnosis compared to those with idiopathic causes (64% vs. 14%)). Panhypopituitarism (defined as deficiencies in ≥3 pituitary hormones) was observed in most patients with an oncologic diagnosis before the start of anti-tumor therapy (*n* = 13). Of interest, patient No. 4, who initially presented with CDI, had an apparent ‘resolution’ of CDI and was able to stop minirin due to the emergence of AI during FU. After the completion of the anti-tumor therapy (chemo-irradiation in GCT, chemotherapy in LCH), the hormonal profile remained stable until the last FU in eight patients; pituitary hormone deficiencies improved in five patients and they worsened in one patient. Among patients with idiopathic PST, the majority of them had stable hormonal profiles during the course of their FU (*n* = 11), two had progressive endocrinopathies, and one showed the regression of the endocrinopathy.

For those presenting with visual problems, we observed a slightly greater number of patients in the neoplastic group (*n* = 5) when compared with the idiopathic group (*n* = 4), but the difference was not clinically significant (*p* > 0.05).

### 3.5. PET-MRI as a Complementary Tool in PST Diagnostics

As an exploratory, proof-of-principle analysis, we incorporated PET-MRI in the evaluation of two patients with PST (Figure 2). Patient No. 28 was a 14-year-old boy who presented with CDI, GHD, HH and AI, with MRI showing PST of 6.6 mm and normal tumor markers. The PET uptake of pituitary tissue at diagnosis showed SUVmax of 9.6, and a subsequent biopsy confirmed the diagnosis of germinoma. Upon completion of chemotherapy, an interval PET scan showed a reduction in SUVmax to 2.8. Patient No. 12 was a 7-year-old girl who presented with CDI and GHD. The growth velocity was decreased to 2.5 cm/year at presentation and her MRI revealed PST of 4 mm. Despite FU for over 2 years, the family refused a CSF examination and biopsy, thus representing a dilemma between the need for GH replacement and the concern about a neoplasm. She eventually underwent a PET-MRI scan, which revealed an SUVmax of 2.6, which was within normal limits. The patient was then counselled on the likelihood of a non-neoplastic cause and GH replacement was commenced.

## 4. Discussion

One of the strengths of our study is its basis on a sizable cohort of patients with endocrine dysfunction and with MRI performed in three pediatric units, which allowed for unbiased subject inclusion with an adequate duration of FU. Additionally, our cohort was uniquely of Chinese ethnicity, adding to the literature data based on a diversity of populations. Neoplastic causes accounted for 50% of the PST in our cohort, which is similar to that reported in the literature (45.5%) based on the summary of a largely Western population (*n* = 684 from 11 studies, 1 study with 48 patients from Asian region) [27]. Compared with the reported pooled experience (13.1% and 16.1% respectively), we saw a higher rate of GCT (46.4%) and lower rate of LCH (3.6%). The only Asian study from this combined analysis was by Liu et al., based on a single-institution experience from Taiwan—among children with DI with/without PST, 32.2% and 19.4% had GCT and LCH, respectively [35]. Nevertheless, only half of these patients with GCT/LCH had PST and the criteria for diagnosis were not specified, rendering such evidence difficult to apply for decision making in the context of children with pituitary stalk anomalies and endocrinopathies. Overall, the observed larger proportion of patients with CNS GCT in our cohort is consistent with the higher incidence of this condition in Chinese and Asians [36,37]. With almost half of our patients suffering from GCTs, the upfront testing of tumor markers and subsequent surveillance in patients with presumed idiopathic disease are essential.

Previous studies did not evaluate predictors of oncologic etiologies in patients with PST. Based on our findings, males with PST were more likely have neoplastic causes, and this is consistent with another study from Korea by Yoon et al. [14]. Additional predictors included severe stalk thickening and CDI with/without additional pituitary hormonal deficiencies. The proposed NICE algorithm takes into account the degree of PST, hormonal disturbances, the evolution of the hormonal profile, radiological features, tumor marker levels and vision for the formulation of the management plan. However, the NICE guidelines do not account for the higher incidence of GCTs in Asians and thus the discordance in risk by sex. In addition, we opine that the guidelines remain relatively conservative when applied to our local context, as surveillance is only recommended after unrevealing second-line investigations (i.e., CSF examination ± whole-body PET for LCH), even in patients with severe PST or a combination of PST and CDI/evolving hormonal deficiencies. Based on our results, we suggest that in ethnic groups where the incidence of CNS GCT is higher, biopsy should be offered after non-diagnostic second-line investigations for patients with PST who (1) are male, (2) have severe stalk thickening (≥6.5 mm) or (3) present with CDI. However, it should be highlighted that even with the histological diagnosis of ‘lymphocytic hypophysitis’, the ultimate underlying diagnosis could still be GCT, as illustrated by three of our patients in this cohort. Ongoing monitoring and appropriate counselling is needed in this subset of patients. In a situation where a decision remains difficult, additional anterior hormonal deficiencies and/or the evolution of hormonal deficiencies should be considered as pressing indicators for tissue sampling. On the other hand, a more conservative approach with serial biochemical and radiological monitoring can be advised for female patients or those presenting with CPP, where PST is more likely due to non-oncologic causes. In such a low-risk group, an annual MRI for three years may be considered before discharging the patient from routine neuroimaging, in the absence of evolving endocrinologic, neurological or visual symptoms; meanwhile, a CSF examination for tumor markers may be considered at the diagnosis of stalk thickening after thorough counseling. The timely discussion of appropriate diagnostic procedures will mitigate the delay in diagnosis and prevent further tumor-related complications, such as hydrocephalus, metastasis and worsening hypopituitarism. As demonstrated in our experience, the loss of pituitary function can be partially reversed in a subset of patients with CNS GCT after treatment.

On the whole, the diagnosis of childhood brain tumors is more often delayed than in other pediatric cancers. This, as demonstrated in our study, is particularly important for sellar–suprasellar tumors, including GCT, low-grade glioma and craniopharyngioma. For such tumors, symptoms can be insidious, atypical and may even wax and wane, adding to the diagnostic challenges [38]. To shorten the lead time to diagnosis, education starting with the medical curriculum should be undertaken [39].

It may be appreciated that even with the availability of tumor tissue, the interpretation of histology may be hampered by sampling bias, tissue heterogeneity and the prominent inflammatory infiltrates frequently present in CNS GCTs. Recently, there has been increasing interest in the application of 11C-MET PET-MRI as a diagnostic tool in patients with intracranial neoplasms [40,41], including initial experience in patients with CNS GCTs [42,43,44,45,46,47]. Such a methodology offers functional data that complement the anatomical data captured by MRI alone. The potential utility of 11C-MET PET includes differentiation between malignant and non-malignant brain lesions, the identification of hypermetabolic regions within neoplasms to guide sampling and the tracking of the response to therapy, including the delineation between disease progression and pseudoprogression [41]. Its application in two of our patients with PST demonstrates its feasibility and supports further evaluation for its utility in this setting.

The limitations of the study include its retrospective nature, the relatively small cohort size and the possibility of the theoretically delayed presentation of neoplasms in those who were labelled thus far as having idiopathic PST. The patient number did not allow a meaningful multivariate analysis of cancer predictors to be performed. Nonetheless, our study represents the only study on Asian children to date focusing on patients presenting with isolated PST. Within the scope of this study, we have not been able to incorporate novel biomarkers for CNS GCTs, such as CSF and serum microRNAs. Specific microRNA species allow differentiation between patients with and without CNS GCTs [48]. MicroRNA profiling may represent a powerful platform as serial monitoring allows the tracking of treatment responses.

## 5. Conclusions

The risk of oncologic etiologies in children with endocrinopathies as well as PST differs according to ethnicity, and our study offers invaluable data on the rates and predictors of oncologic outcomes in a carefully curated Chinese cohort. CNS GCTs are more prevalent in Asian patients, calling for an ethnicity-specific diagnostic strategy, with a lower threshold for biopsy in patients of male sex, with severe stalk thickening and CDI. Careful scrutinization of sampled tissue is essential, while PET-MRI represents a potentially useful tool to complement existing modalities of investigation. Early diagnosis will allow the better preservation of function.

## Figures and Tables

**Figure 1 cancers-15-03935-f001:**
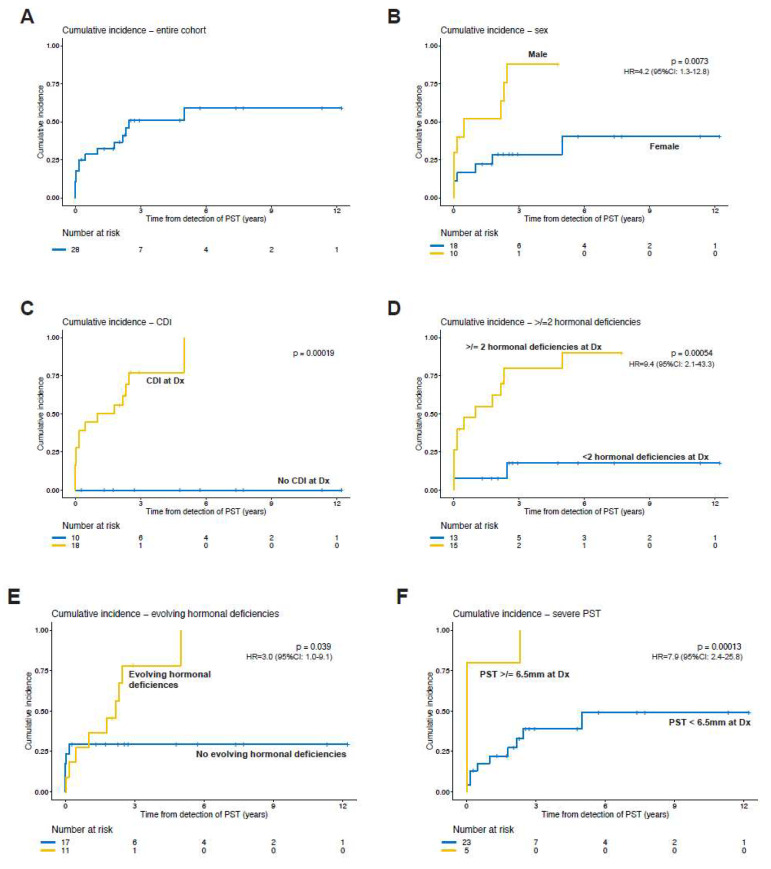
Cumulative incidence of oncologic diagnosis for (**A**) the entire cohort, and by significant predictors, including (**B**) sex, (**C**) central diabetes insipidus (CDI) at diagnosis (Dx), (**D**) 2 or more pituitary hormonal deficiencies at Dx, (**E**) evolving hormonal deficiencies and (**F**) severe pituitary stalk thickening (PST) at Dx. *p*-values calculated by log-rank analysis. Hazard ratios (HR) and 95% confidence intervals (CI) derived from Cox regression (not available for panel (**C**) as no event occurred in the subset with no CDI).

**Figure 2 cancers-15-03935-f002:**
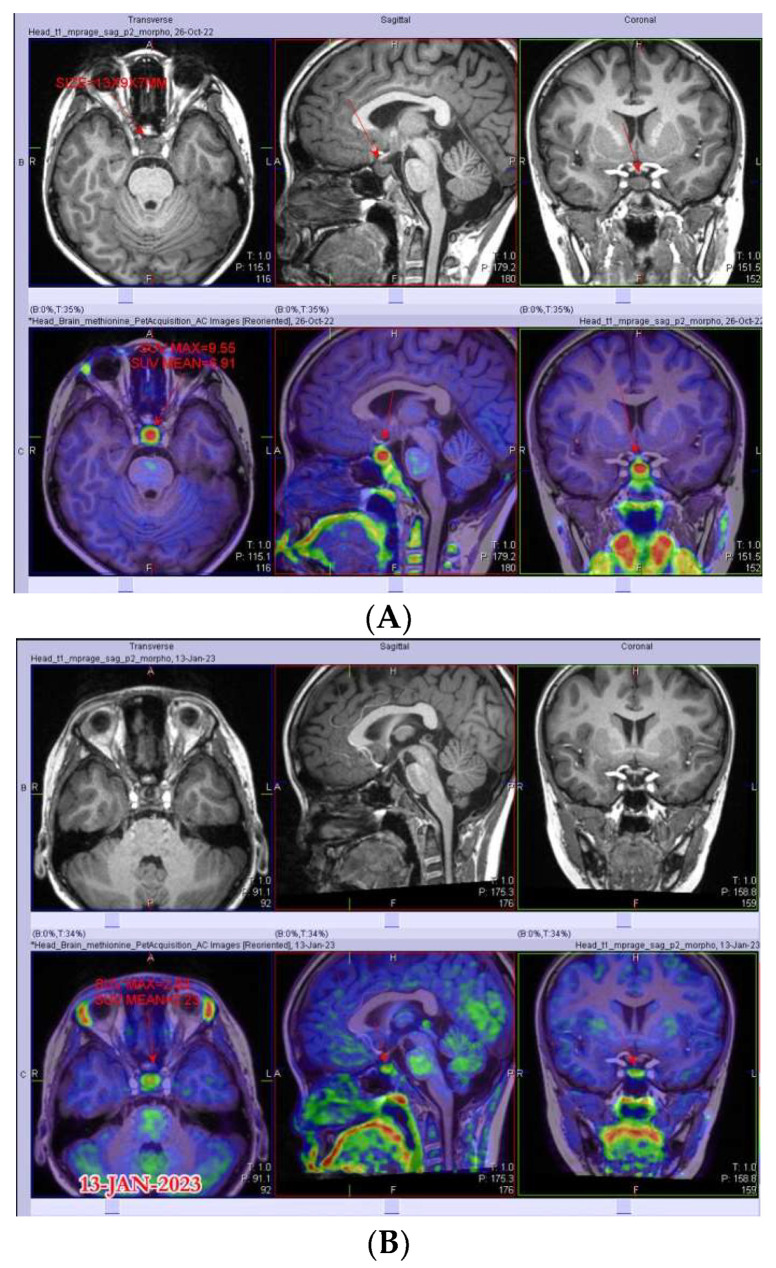
Representative images from C11-methionine PET-MRI on two patients with PST—location of pituitary stock indicated by white circles on composite axial images. Patient No. 28, with biopsy-proven germinoma, with hypermetabolic pituitary stalk (SUVmax 9.6) (**A**), which responded after treatment (SUVmax 2.83) (**B**). Patient No. 12 (**C**), with normal metabolic activity at the stalk region (SUVmax 2.6), who was then followed clinically and radiographically for presumed idiopathic stalk thickening.

**Table 1 cancers-15-03935-t001:** Summary and evolution of endocrinopathies at diagnosis, pre-treatment and latest follow-up among patients with and without known oncologic etiologies.

Patient Number	Final Diagnosis	Sex	Age of Dx of PST (Years)	Growth Hormone			Gonadal Axis			Thyroid Axis			Adrenal Axis			Central Diabetes Insipidus			CPP
Patient with Oncological Diagnosis				At Dx	Pre Tx	Last FU	At Dx	Pre Tx	Last FU	At Dx	Pre Tx	Last FU	At Dx	Pre Tx	Last FU	At Dx	Pre Tx	Last FU	At Dx
1	Germinoma	F	10.3	↓	↓	↓	Unknown	↓	Normal	Normal	Normal	Normal	Normal	↓	↓	Yes	Yes	Yes	No
3	Germinoma	F	5.9	↓	↓	↓	Unknown	Unknown	Normal	Normal	↓	↓	Normal	↓	↓	Yes	Yes	Yes	No
4	Germinoma	M	16.5	↓	↓	Unknown	Normal	↓	↓	Normal	↓	↓	Normal	↓	↓	Yes	Yes	Yes	No
8	NGGCT	M	8.2	↓	↓	↓	↓	↓	↓	Normal	↓	↓	↓	↓	↓	Yes	Yes	Yes	No
9	LCH	F	5.2	Normal	Normal	↓	Normal	Normal	Normal	Normal	Normal	↓	Normal	Normal	Normal	Yes	Yes	Yes	No
10	Germinoma	F	9.4	↓	↓	↓	Unknown	Unknown	Unknown	↓	↓	↓	Normal	↓	↓	Yes	Yes	Yes	No
13	Germinoma	M	14.6	Normal	Unknown	Unknown	Normal	↓	↓	Normal	↓	↓	Normal	↓	↓	Yes	Yes	Yes	No
14	NGGCT	M	9.5	Normal	Normal	↓	Unknown	↓	Unknown	↓	↓	↓	↓	↓	↓	Yes	Yes	Yes	No
16	Germinoma	M	13.4	↓	↓	↓	↓	↓	Normal	↓	↓	↓	↓	↓	↓	Yes	Yes	Yes	No
20	Germinoma	F	12	↓	↓	Unknown	Normal	↓	↓	↓	↓	↓	Normal	↓	↓	Yes	Yes	Yes	No
24	Germinoma	M	10.9	↓	↓	↓	Normal	Normal	Normal	↓	↓	↓	↓	↓	Normal	Yes	Yes	Yes	No
25	Germinoma	F	12.2	↓	↓	Unknown	↓	↓	↓	↓	↓	↓	↓	↓	↓	Yes	Yes	Yes	No
26	Germinoma	M	15.4	↓	↓	Normal	Normal	↓	↓	Normal	↓	↓	Normal	↓	↓	Yes	Yes	Yes	No
28	Germinoma	M	14.1	↓	↓	↓	↓	↓	Normal	Normal	Normal	Normal	↓	↓	↓	Yes	Yes	Yes	No
**Patient Number**	**Final Diagnosis**	**Sex**	**Age of Dx of PST (Years)**	**Growth Hormone**			**Gonadal Axis**			**Thyroid Axis**			**Adrenal Axis**			**Central Diabetes Insipidus**			**CPP**
**Patient with Non-Oncological Diagnosis**				**At Dx**		**Last FU**	**At Dx**		**Last FU**	**At Dx**		**Last FU**	**At Dx**		**Last FU**	**At Dx**		**Last FU**	**At Dx**
2	Idiopathic	F	10.9	Normal		Normal	Normal		Normal	Normal		Normal	Normal		Normal	Yes		Yes	No
5	Idiopathic	M	15.6	↓		↓	↓		↓	Normal		Normal	Normal		Normal	No		No	No
6	Idiopathic	F	10.4	Normal		Normal	Normal		Normal	Normal		Normal	Normal		Normal	No		No	Yes
7	Idiopathic	F	15	↓		↓	↓		↓	Normal		Normal	Normal		Normal	Yes		Yes	No
11	Idiopathic	F	3.8	Normal		↓	Normal		Normal	Normal		Normal	Normal		Normal	Yes		Yes	No
12	Idiopathic	F	5.6	Unknown		↓	Normal		Normal	Normal		Normal	Normal		Normal	Yes		Yes	No
15	Idiopathic	F	13.8	Normal		Normal	Normal		Normal	Normal		Normal	↓		Normal	No		No	No
17	Idiopathic	F	8.8	Normal		Normal	Normal		Normal	Normal		Normal	Normal		Normal	No		No	Yes
18	Idiopathic	F	11.6	Normal		Normal	Normal		Normal	Normal		Normal	Normal		Normal	No		No	Yes
19	Idiopathic	F	9.7	Normal		Normal	Normal		Normal	Normal		Normal	Normal		Normal	No		No	Yes
21	Idiopathic	F	9.7	Normal		Normal	Normal		Normal	Normal		Normal	Normal		Normal	No		No	Yes
22	Idiopathic	F	7.3	Normal		Normal	Normal		Normal	Normal		Normal	Normal		Normal	No		No	Yes
23	Idiopathic	F	16.2	↓		↓	↓		↓	Normal		Normal	Normal		Normal	No		No	No
27	Idiopathic	M	13.5	Normal		Normal	Normal		Normal	Normal		Normal	Normal		Normal	No		No	Yes

## Data Availability

The data presented in this study are available in this article.

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
