# Peer review of "Incidence and Predictors for Oncologic Etiologies in Chinese Children with Pituitary Stalk Thickening"

_cancers, 2023, doi:10.3390/cancers15153935_

Round 1

Reviewer 1 Report

The authors are reporting the result of a retrospective study of pediatric patients with pituitary stalk thickening (PST) at 3 hospitals in Hong Kong. They identified 28 patients meeting eligibility criteria of endocrinopathy and PST with a follow-up (FU) duration of at least 2 years in patients without oncologic diagnosis. The aim of this study was to describe the natural history of PST, to identify predictors of oncologic etiologies and assess the predictive value of 11C-methionine (MET) PET-MRI studies in this context (used in 2 patients). With a median FU of 4.7 years, 14 patients were diagnosed with oncologic condition, including 13 with a diagnosis of germ cell tumor. 

Specific comments:

- The authors mentioned that they adopted definitions per the NICE guideline in this study. NICE does not recommend contrast administration in the MRI evaluation of PST. Did the authors look at contrast enhancement in their series (no data on this aspect in reference 27 from NICE). This is important, as some enhancing lesion tend to lose contrast other time, and this may be considered as an MRI predictor of non neoplastic lesion.

- Line 157: the authors describe endocrinopathies and mention that the initial symptoms were polyuria and polydipsia in 17 patients. Can they provide a duration of polyuria and polydipsia for these 17 patients?

- Section Result: How often were MRI scan performed in this experience. It would be interesting to have information on the growth of PST overtime and know if growth is predictive of oncologic etiology. 

- Line 176: one can read: "Patient No. 8 had an initial diagnosis of hypophysitis, and underwent 3 biopsies over the course of 22 months before the diagnosis of NGGCT was made." This could deserve more comments: what did the 2 first biopsies show? Why was a biopsy done when the level of HCG increased in the serum, as it was diagnostic for GCT at that time? Was staining for CD30 repeated on the biopsy 1 and 2 when biopsy number 3 showed embryonal carcinoma?

- In the discussion, the authors state "males with PST were more likely have neoplastic causes, likely attributed to the male predominance in GCTs". This is not exact for GCT occurring in the suprasellar region that have a balanced male/female ratio (the male predominance ins essentially in pineal and basal ganglia GCT.

- In the discussion, the author shoudl make recommendation based on their experience on the timing of MRI scans. It woudl also be interesting to comment on the necessity (or not) of MRI monitoring in patients without DI (no tumor in this series) and on the value of CSF studies

Reviewer 2 Report

I comment the authors on this interesting study. This is an interesting study and will advance our knowledge and help improve time to diagnosis.

Comments/suggestions:

1-      Remove the parts about MET PET. It does not add to the paper, and it was done for only two patients. If the authors insist on keeping it, I am fine, but they need to tone this part down. It is over emphasized.

2-      On page 9 the authors cite reference 35 (the Italian study by Maghnie et al in NEJM). They discuss this study as it is a literature review that included Liu et al study. Please check this reference again and this paragraph (lines 261 till 272).

3-      The authors need to discuss delayed diagnosis in CNS tumors in general then common tumors in supra sellar area: craniopharyngioma and GCT as they did but also low-grade gliomas. Discuss the need to include such issues as this study in medical education curricula. See the following studies to help the authors (PMID: 25742877 and PMID: 31691058)

4-      The authors need to discuss how in this location other symptoms can occur such as anorexia from LGG and hyperphagia from craniopharyngioma (PMID: 35149436).

5-      The table can be improved. Quality wise and not information.

6-      Pleas check other references to make sure the authors cite them correctly. See comment 2 again.

Reviewer 3 Report

Thank you for giving me the opportunity to read this interesting manuscrip. I only have few comments, mainly concerniny statistical methodology:

P5, table 1: font is very small to read

How did you calculate the median follow-up time, did you use the reverse Kaplan-Meier method?

How did you calculate „The median time from appearance of first symptom to diagnosis of GCT/LCH was 1.3 years (range: 0.5-5.4).“ and other median times?

It does not seem, as if you used Kaplan-Meier estmates?

Reviewer 4 Report

Several points need to be revised:

- Lines 74-76: "As an exploratory study, the utility of 11C-methionine PET-MRI in the context of PST was also evaluated" It not clear what is the purpose of this paper. Improve the introduction part.

- Lines 185-187: "Her serum and CSF tumour markers were again all along normal. A biopsy was performed, which was initially misdiagnosed as lymphocytic hypophysitis. However, on further pathology review, germinoma cells were identified". Why do the authors highlight this part? Discuss later this point.

- Table 1 is very hard to read it. Try to merge it or to reduce it.

- Lines 269-272: "All in all, the observed higher proportion of patients with CNS GCT... idiopathic disease are essential"

- Lines 277-279: "Additional predictors included severe.." In the discussion section, some very important refereces must be considered. Look at these ones: -- doi: 10.4103/sni.sni_398_17 -- doi: 10.1016/j.clineuro.2023.107830  -- doi: 10.7417/CT.2022.2388

- Add a "limitations section" to this paper.

- Conclusion is very short. Improve it. Why do authors write this paper? What they want to add new to the current literature?

Minor editing of English language required

Round 2

Reviewer 4 Report

Good

Minor editing of English language required